# Exploring the eating experience of a pneumatically-driven edible robot: Perception, taste, and texture

**Yoshihiro Nakata**[1]*, **Midori Ban**[2], **Ren Yamaki**[2], **Kazuya Horibe**[2], **Hideyuki Takahashi**[2], **Hiroshi Ishiguro**[2]

**1** Graduate School of Informatics and Engineering, The University of Electro-Communications, Chofu, Tokyo, Japan, **2** Graduate School of Engineering Science, Osaka University, Toyonaka, Osaka, Japan

* ynakata@uec.ac.jp

**Data Availability Statement:** Data are available from the following URL and DOI: URL: https://figshare.com/articles/dataset/25046240 DOI: 10.6084/m9.figshare.25046240.

## Abstract

This study investigated the effects of animated food consumption on human psychology. We developed a movable, edible robot and evaluated the participants' impressions induced by the visualization of its movements and eating of the robot. Although several types of edible robots have been developed, to the best of our knowledge, the psychological effects associated with the eating of a robot have not been investigated. We developed a pneumatically driven edible robot using gelatin and sugar. We examined its perceived appearance and the participants' impressions when it was eaten. In the robot-eating experiment, we evaluated two conditions: one in which the robot was moved and one in which it was stationary. Our results showed that participants perceived the moving robot differently from the stationary robot, leading to varied perceptions, when consuming it. Additionally, we observed a difference in perceived texture when the robot was bitten and chewed under the two conditions. These findings provide valuable insights into the practical applications of edible robots in various contexts, such as the medical field and culinary entertainment.

## Introduction

Eating, as a means of nutrition, is an integral part of human life, intended to sustain life. However, it is also a social and cultural activity [1, 2]. Occasionally, it involves the act of taking the life of another living creature. Thus, exploring the relationship between people and living creatures related to food consumption is an interesting study subject [3]. For example, morality and associated feelings of guilt, which results from eating living creatures (cf. meat paradox), vary from country to country [4]. A unique food culture in which people eat live fish and shellfish exists in Japan termed "Odorigui" [5]. However, scientific experiments cannot easily be conducted using living creatures to explore the psychological and cognitive mechanisms governing these cultures. Although experiments can be conducted using living creatures, controlling their appearance and movements is challenging when conducting controlled experiments.

Because of the challenges of investigating consumption behaviors using living creatures, we introduced the concept of human–edible robot interaction (HERI). This study elucidates the interaction between humans and edible robots by offering a controlled environment to

**Funding:** H. I. Grant Number JPMJMS2011 Moonshot R&D, Japan Science and Technology Agency Grant Number JP19H05693 Grant-in-Aid for Scientific Research on Innovative Areas (Research in a proposed research area), Japan Society for the Promotion of Science Y. N. Grant Number XC2022008 Support for External Funds Acquisition by Early Career Scientists, The University of Electro-Communications The funders had no role in study design, data collection and analysis, decision to publish, or preparation of the manuscript.

**Competing interests:** The authors have declared that no competing interests exist.

investigate human psychology when engaging with robots that are consumable even in motion. We recognize that the full replication of animality in edible forms remains a distant goal for HERI. Therefore, this study focuses on the immediate psychological responses elicited by the consumption of movable robots, rather than mimicking living organisms. Our primary aim in this initial phase is to investigate the psychological and cognitive effects that arise from such novel interactions.

Exiting studies on edible robots have focused on developing robotic elements [6–9] with low environmental loads and medical robots [10] with little negative impacts on humans. However, these studies have not considered the taste and eating sensations of edible robotic parts. For example, edible actuators [7] using gelatin–glycerol compounds have been molded and dried to increase their strength. However, they were considered hard and could not easily be bitten and chewed. Medical robots [10] that operate inside the human body have been fabricated to be easily swallowed. In contrast, we developed edible robots by considering their taste and eating sensations and experimentally investigated the psychological and cognitive effects of the consumption of these moving robots on participants.

Given the novelty of this research direction, our study primarily stands as exploratory, laying the foundational insights and understanding in the field of HERI. Herein, we examined the development of an edible robot whose moving part can be bitten and chewed and evaluated the participants' impressions on this act. First, we developed a pneumatically driven robot with a hardness suitable for biting and chewing using gelatin and sugar. In a preliminary study, we investigated the visual impression impacted by the movement of the developed edible robot. In the main study, we experimented with an edible robot under two operating conditions: without robot movement (stationary condition) and with robot movement (movement condition). After parts of the robot were consumed by participants, we evaluated their responses to the animateness, deliciousness (taste and texture), appetite, and sense of guilt for eating parts of the robot. These factors are considered essential evaluation items in HERI.

The contributions of this study are as follows:

- We developed a movable, edible robot intended for consumption. We designed and built a pneumatically driven edible robot using gelatin and sugar, incorporating flavor and a soft texture. This represents a novel approach, as previous studies on edible robotics did not focus on the actual eating experience [6–9].

- Experiment 1. We evaluated participants' impressions of the robot solely based on the visualization of its movements, without actually eating it. This provided insights into the effect of the perceived animateness of the robot on the hypothetical eating experience.

- Experiment 2. In this robot-eating experiment, by comparing two conditions, one in which the robot was moved and one in a stationary state, we identified differences in the perception and taste of the participants concerning consuming the animated food and differences in the perceived texture when biting and chewing the robot.

## Edible robot

This section describes the design, development, control method, and control system of the robot. The material used for the edible part of the robot and its characteristics were introduced. This study focused on manufacturing a robot that can be driven using edible materials. We chose a pneumatic approach for its drivability, given the harmlessness of air when ingested. Considering the actual eating while it moves, we designed the robot in a stick shape,

a simple form. To conduct an experiment in which the edible part of the robot was eaten, materials were selected accordingly, and the robot was developed to satisfy the following conditions:

- Edible parts comprise materials that are safe to consume

- The edible part is not damaged when the robot moves but can be bitten and chewed as food

Gelatin was used to create a structure, which embeds air chambers, and the robot was driven by deformation resulting from changes in air pressure. Several materials were added to gelatin to increase the strength of the material and add flavor.

## Structure

Pneumatically driven soft actuators use the deformation caused by balloon-like structures made of flexible materials, which are inflated to drive. Soft pneumatic actuators primarily comprise two types those that use only a single material [11, 12] and others that combine a flexible with a hard-to-stretch material, such as a wire [13–15]. Notable, materials with different hardness values can be created by varying the amount and combination of materials. However, for simplicity, we created an edible part using only a single hardness material.

Fig 1 shows a three-dimensional (3-D) computer-aided design model of the edible robot and its internal structure. It consists of an edible part made of material will describe in the "Edible material" section and a base component, which is connected to an air tube to supply air to the edible part. As a pioneering endeavor in the field, we provided participants with the unique experience of consuming a robot while moving it. Thus, it was not feasible to cut the robot into smaller pieces before eating. Therefore, we had to design a shape and size that could enable the robot to be placed directly into the mouth. This led to stick-shaped robots. Additionally, owing to the constraint of the size of the robot that could fit in the mouth, the edible part was designed to have only two air chambers.

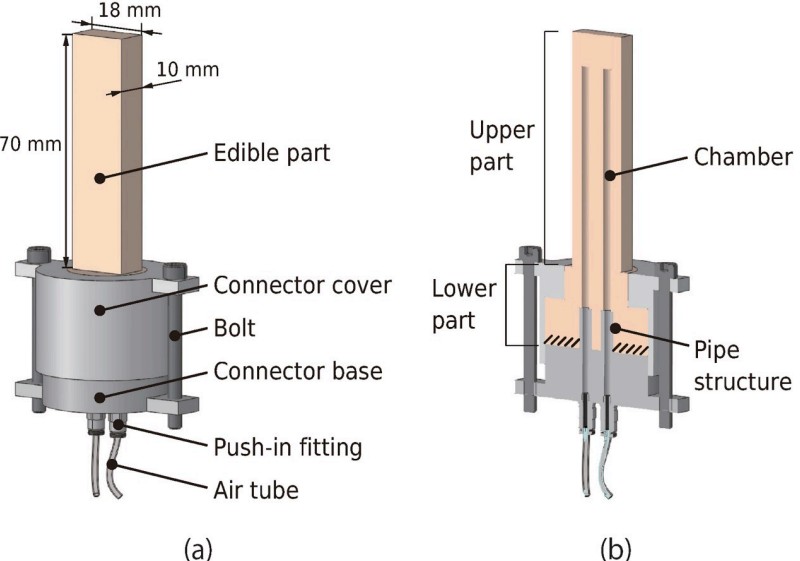

**Fig 1. Three-dimensional computer-aided design model of the edible robot.** (a) Edible robot and its (b) internal structure.

One of the essential steps in the development of pneumatically driven soft actuators is the prevention of air leakage at the connection point of the air-supply tube. Yirmibesoglu et al. [12] fixed the connecting part between a soft actuator and tube using an adhesive to prevent air leakage. However, connecting an edible material to a urethane tube with glue is challenging. We believe that glue should be avoided for health and safety reasons. Therefore, we adopted a method to prevent air leakage by tightly sealing the edible part of the connector bases, where two pipe structures protruded from the upper surface of the connector base. A hole in the pipe center extended to the under-surface of the connector base, where the air was supplied to the edible part through a connecting tube, and a one-touch joint was used to connect the tubes. Pipes were inserted into each air chamber, and the edible part was set. The edible upper and lower parts were rectangular and cylindrical, respectively, and the lower part was radially thicker than the upper part. The lower part of the edible part was sandwiched between the connector base and cover, such that the hatched area in the figure adhered to the connector base, to prevent air leakage. Although the connector base and cover apply vertical pressure to the lower part of the edible part, the edible part itself is not easily broken, as the force is applied only to its flat surface.

## Operating principle

The thickness of the walls surrounding the air chamber can be assumed to be non-uniform. In this case, when the air chamber expands owing to the air supply, the thinner wall experiences greater expansion, thus creating a difference in the wall elongation in the longitudinal direction of the air chamber. This part then bends toward the smaller elongation side. The edible part of the developed robot was designed such that the wall between the two chambers is thicker than the surrounding wall. As the thicker part of the wall cannot easily be extended when air is supplied to one of the air chambers, the edible part bends in the direction of the other air chamber. When the same amount of air is simultaneously supplied to both air chambers, the edible part extends in the direction of the length of the air chamber. In this experiment, the robot was operated by alternating or simultaneous air supply to (and exhaust from) the two chambers.

## Edible material

In the development of our pneumatically driven edible robot, one of the crucial challenges was to identify a material that enabled the necessary movement and inflation and was also edible and had a texture suitable for consumption. The material needed to have sufficient elasticity to withstand the stresses of pneumatic action without tearing, while also being chewable and edible as a food when consumed. This section describes our selection and reasoning for the edible materials used in the robot.

An example of the material used for pneumatically driven soft actuators is silicone rubber [16]. To realize a pneumatically driven edible robot, elastic edible materials should be used. Gelatin and agar are examples of elastic-gelling foods. In their study on the chewing patterns of soft foods, Arai et al. [17] investigated the properties of gelatin and agar. They showed that gelatin has a higher rupture stress than agar when both exhibit the same hardness values during shear tests. Therefore, in this study, gelatin was used as an elastic- and fracture-resistant material.

The strength and plasticity of gelatin changes when it is combined with other materials. For example, when it is combined with glycerol, it produces soft drug capsules, and when it is combined with sugar or syrup, it produces "gummy" forms. Shintake et al. [7] used gelatin–glycerol material for pneumatic actuators. By adding glycerol and drying the material, its strength

increased. However, in this study, we opted for a mixture of gelatin and sugar, akin to commercially available gummies, to facilitate easy consumption, contributing a moisturizing effect and enhancing flavor [18]. We specifically chose not to dry our material.

We used calcium carbonate and 100% apple juice as solvents, in addition to gelatin and sugar. Calcium carbonate is used as a rubber-reinforcing agent in industrial applications. In this case, we used it as a food additive to reinforce the particles of the edible part. We used as much gelatin as possible to render it elastic and adjusted the amounts such that all of the gelatin and sugar dissolved in the solvent. After several trials, the optimal ratio of gelatin:juice:sugar:calcium carbonate was set to 25:100:30:1. We poured the mixture into a mold (will describe in the "Robot development" section) and refrigerated it for 12 h to ensure solidification (see S1 Fig).

## Tensile-strength testing

To ensure that the edible robot can function optimally without tearing during pneumatic action, we used tensile-strength material testing. This testing compares our selected edible material with existing materials in terms of their tensile strength, especially focusing on the effects of calcium carbonate addition. Furthermore, a reproduction of the gelatin-based-pneumatic-actuator material developed by Shintake et al. [7] was made, whereafter we measured its characteristics and compared the results with those of our robot. The material was prepared using gelatin:glycerol:water = 1:1:8 based on the results of previous studies and was dried at room temperature (25˚C for 2 days). In total, we used three materials: reproduced gelatin and glycerol (1:1) (Gelatin/Glycerol (1:1)), gelatin and sucrose (sugar) with calcium carbonate (our material, Gelatin/Sucrose with $CaCO_3$), and gelatin with sucrose and without calcium carbonate (Gelatin/Sucrose without $CaCO_3$).

A microforce tensile-strength testing machine (Tytron250, MTS Systems Corporation) was used for the tests (see Fig 2(a)). The specimen was cut into a dog-bone shape, as shown in Fig 2(b), and set in the machine, as shown in Fig 2(c). The thickness of the specimen was 1 mm. Both ends of the specimens were elongated and continuously loaded until rupture. The tensile speed was set to 0.01 mm/s. The measurement was performed only once for each material.

To obtain the mechanical properties of each material, Young's moduli and tensile strengths were calculated from the measured data using Yeoh's hyperelastic-material model [19]. Model assumptions and calculations were conducted based on the literature [7].

Fig 2(d) shows the tensile testing results. The material constants $C_1$, $C_2$, and $C_3$ of Yeoh's model were determined to fit the model to the measurement results. The constants for each material are listed in Table 1.

The Young's modulus of material $E$, considering the consistency condition, was obtained as follows,

$$E = 2\mu(1 + v), \tag{1}$$

where $v$ is Poisson's ratio of the material and $\mu$ is the shear modulus, which is obtained according to

$$\mu = 2C_1. \tag{2}$$

The Poisson's ratio was approximated as 0.49 in this study.

The Young's modulus, tensile strength, and elongation at break values are summarized in Table 2. The results indicate that all these properties improved with the addition of calcium carbonate in the case of gelatin–sucrose, compared with the case in which no calcium carbonate was present. This suggests that the addition of calcium carbonate to the gelatin–sucrose

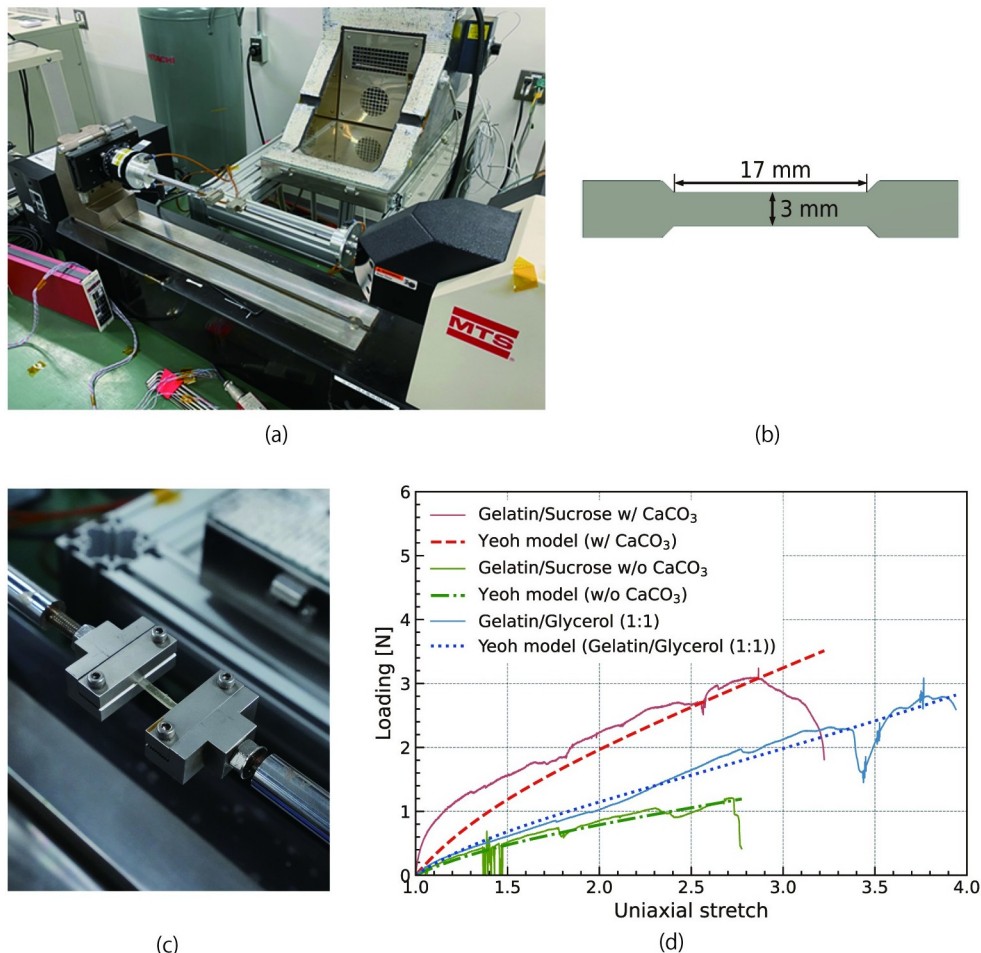

**Fig 2. Tensile-strength testing.** (a) Microforce tensile-strength testing machine, (b) dimensions of the specimen (dog-bone shaped), (c) an enlarged photograph of the test specimen on the machine, and (d) results of tensile-strength testing.

**Table 1. Material constants.**

| Constant [MPa] | Gelatin/Sucrose with CaCO$_3$ | Gelatin/Sucrose without CaCO$_3$ | Gelatin/Glycerol (1:1) |
|:---:|:---:|:---:|:---:|
| $C_1$ | $1.87 \times 10^{-1}$ | $7.52 \times 10^{-2}$ | $1.07 \times 10^{-1}$ |
| $C_2$ | $3.31 \times 10^{-21}$ | $1.74 \times 10^{-22}$ | $5.41 \times 10^{-4}$ |
| $C_3$ | $6.57 \times 10^{-21}$ | $8.32 \times 10^{-23}$ | $1.84 \times 10^{-29}$ |

**Table 2. Material properties.**

| Property | Gelatin/Sucrose with CaCO$_3$ | Gelatin/Sucrose without CaCO$_3$ | Gelatin/Glycerol (1:1) |
|:---:|:---:|:---:|:---:|
| Young's modulus [MPa] | 1.12 | 0.45 | 0.64 |
| Tensile strength [MPa] | 0.60 | 0.14 | 0.86 |
| Elongation at break [%] | 222.4 | 177.5 | 294.1 |

material is expected to render the edible parts of the robot less prone to breakage. The reproduced material exhibited a large elongation at break; however, Young's modulus and tensile strength were smaller than those in the original study [7]. We believe that the drying conditions were different and that the material was not sufficiently dried. Thus, the properties of our material could not be compared with those of the previous study by simply using the data from this experiment.

## Hardness measurements

The texture of the edible robot should enable movement and render it palatable. The hardness of the material was measured to determine whether the assumed edible part was edible. The hardness of the material was compared with that of commercially available gummies and the reproduced pneumatic actuator using gelatin material developed by Shintake et al. [7]. We fabricated the reproduced material using the same procedure for the material used in the tensile-strength testing. Notably, the tensile-strength testing results showed that the reproduced material was softer than that proposed by Shintake et al. [7]. A durometer (C-type [20], KOBUN-SHI KEIKI CO., LTD.), which was vertically pressed into the test specimen, was used to measure the Asker C hardness. Fig 3(a)–3(d) show the test specimens of the developed edible part, the reproduction of the material developed by Shintake et al., and the two types of commercially available gummies (Lotte Fit's BIG gummy soft and hard types, LOTTE CO., LTD). The measurement was performed five times for each specimen, and the presented results are the averages of the measured values.

The results of the hardness measurements are presented in Fig 3(e). The hardness of the edible part material developed in this study was similar to that of the hard-type gummy. The hardness of the reproduced material by Shintake et al. was almost twice that of our edible part. This appears to be owing to the effects of glycerol and drying. Material hardness is an essential parameter for realizing a practical robotic actuator. However, an extremely high hardness renders the object unsuitable for eating.

## Robot development

Fig 4 shows a photograph of the mold used to create the edible part. The inner surface of the mold in contact with the ingredients was treated with Teflon. The mold was divided into two parts: one for forming the outer shape of the edible part, which is divided into two parts (Fig 4(a-i)), and the other for forming the air chamber (Fig 4(a-ii)). Fig 4(a-iii) shows the base component, which maintains the position of the two outer molds. Fig 4(b) shows the partially assembled molds. The edible material was molded by being poured into these molds.

Fig 5(a) shows a photograph of an edible part attached to the connector to supply air to the air chamber. A 3-D printed handle covered the connector to facilitate holding during the experiment (see Fig 5(b)).

## Control system

The configuration of the control system of the edible robot is shown in Fig 6. The devices used in the control system are presented in Table 3. The developed edible robot was operated by controlling the timing and cycle of the air supply and exhaust to two air chambers inside the edible part. The microcomputer module turns the 3-port solenoid valves on and off via amplifier circuits to switch the air supply and exhaust states.

Air was supplied by an oil-free air compressor that contained no suspended oil, and a regulator adjusted the pressure. The air was passed through an air filter to remove impurities before it was supplied to the edible part. The air filter had a filtration degree of 0.3 $\mu$m.

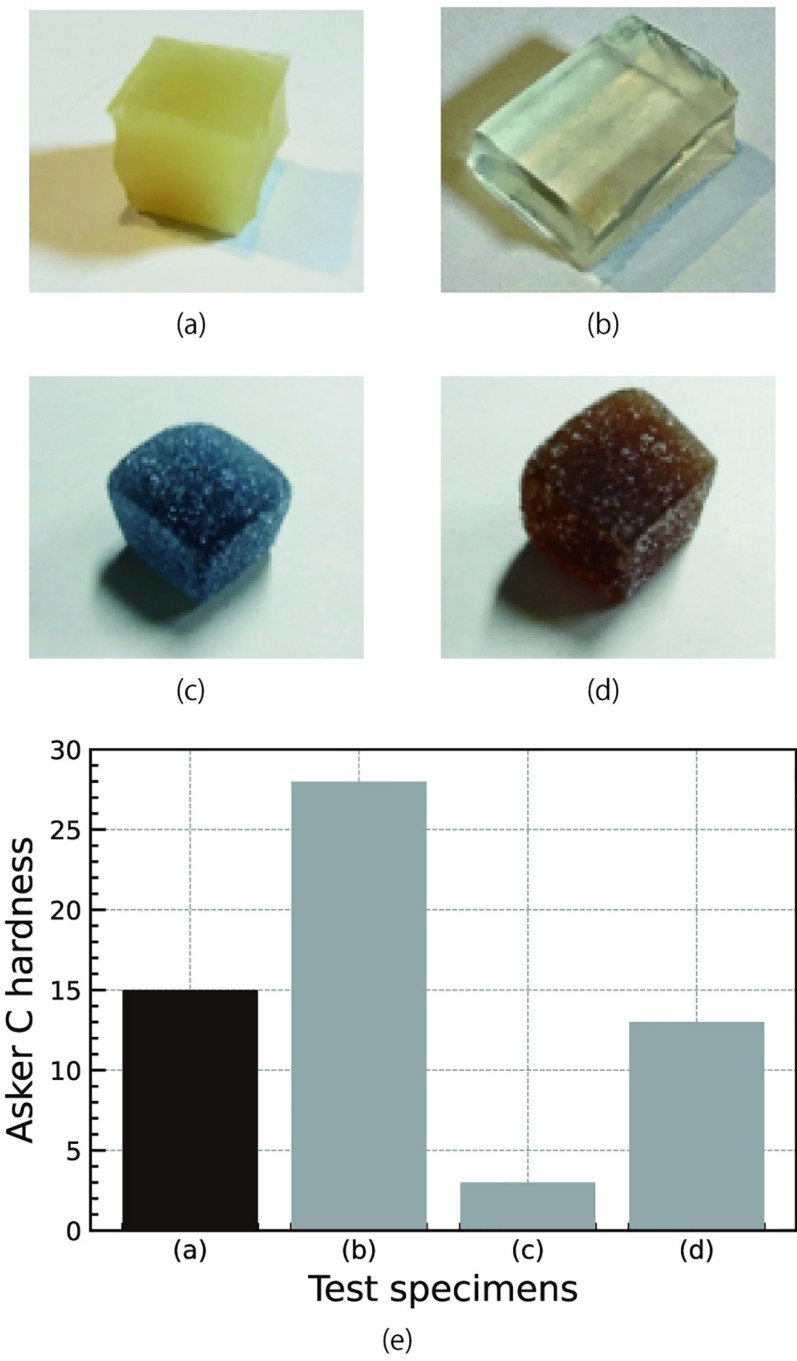

**Fig 3. Test specimens and hardness measurements.** (a) Material of an edible part of the developed edible robot, (b) reproduction of gelatin–glycerol material for an edible robotic actuator made by Shintake et al., (c) soft gummy (Lotte Fit's BIG Gummy soft type, LOTTE CO., LTD), (d) hard gummy (Lotte Fit's BIG Gummy hard type, LOTTE CO., LTD), and (e) comparison of the measured Asker C hardness.

## Experiment 1

In this study, we developed an edible robot system and explored the influence of robot movement on the eating experience. Experiment 1 involved participants watching videos of various robotic movements, with the goal of investigating the impressions these animations evoked,

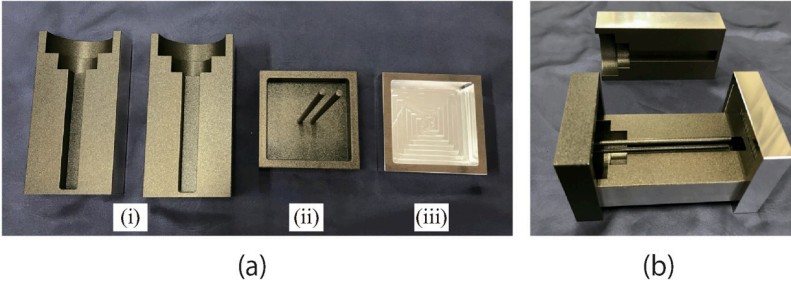

**Fig 4. Molds used to create the edible part.** (a-i) Molds used to form the external shape of the edible part, (a-ii) mold used to form the chambers of the edible part, (a-iii) base component maintaining the position of two molds to form the external shape, and (b) partially assembled molds. When pouring the material, the base part should be positioned at the bottom, and the mold forming the chamber should be inserted after pouring.

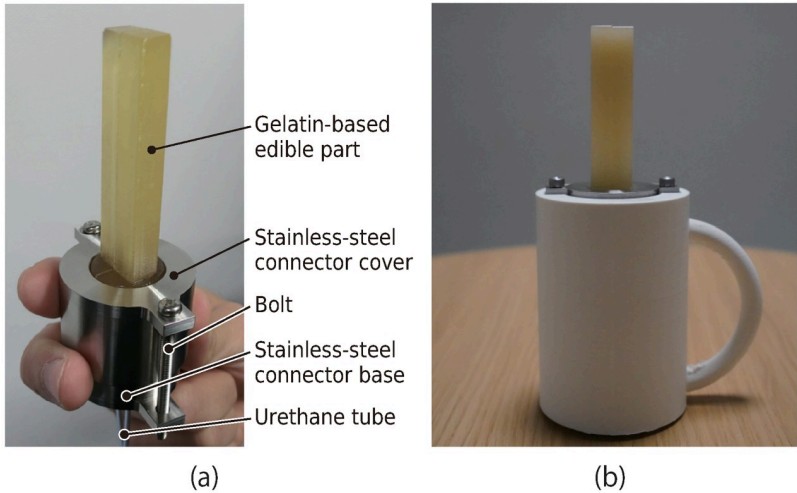

**Fig 5. Edible robot.** (a) Edible part attached to the connector, (b) edible robot with a three-dimensional printed handle.

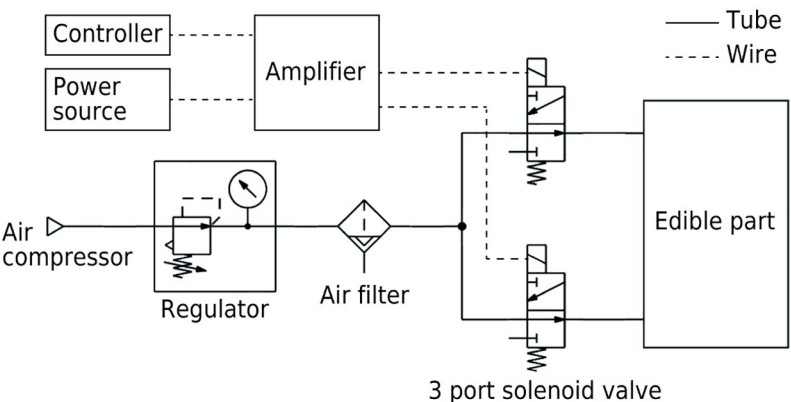

**Fig 6. Control system configuration.**

**Table 3. Devices in the control system.**

| Name | Model number | Manufacture | Specifications |
|---|---|---|---|
| 3 port isolated valve | LVMK207–5J | SMC Corporation | Voltage: 24 V direct current |
| Controller | Mbed LPC1768 | NXP Semiconductors N.V. | |
| Air filter | F8000–20-W-FY | CKD Corporation | Nominal filtration rating: 0.3 $\mu$m |
| Regulator | IR2020–02BG-R | SMC Corporation | Maximum operating pressure: 1 MPa |
| Air compressor | SLP-07EED | ANEST IWATA Corporation | Power source: Single-phase 100 V<br>Motor power: 0.75 kW<br>Working pressure: 0.6–0.8 MPa |

specifically regarding "the perception of an edible robot" and "taste." It is important to note that this phase of the study did not entail actual consumption of the robot, but rather participants' reactions were based solely on visual observation.

## Participants

An online survey was administered to participants registered on the crowd-sourcing service. They completed the survey by accessing the online forms. A total number of 315 participants were enrolled (117 males, 196 females, and two non-respondents; mean age = 39.13 years old, range = 18–49 years old, and SD = 7.63 years old).

## Materials and methods

This study protocol was approved by the Ethics Committee of the Graduate School of Engineering Science at Osaka University (No. R3–18). All participants provided written informed consent before they participated in the experiment. The experiment was performed per the principles and guidelines of the Declaration of Helsinki (1975).

**Robot movement conditions.** Two factors were adjusted to control the movement of the developed edible robot: the timing and cycle of air supply and exhaust to and from the two air chambers in the edible part. For the timing of the air supply and exhaust, we set up two conditions: the first was one air chamber at a time (hereafter, referred to as the "alternating" condition), and the second condition was the cycle of air supply and exhaust in which air was supplied to both air chambers (hereafter referred to as the "simultaneous" condition). In the alternating condition, the robot swings laterally, producing movements reminiscent of spinal flexion. By contrast, in the simultaneous condition, the robot moves up-and-down, generating movements akin to the up-and-down motion of the torso caused by knee flexion and extension. These movement patterns were selected based on the range of movements that the developed edible robot could realistically achieve, given its design and mechanism. Notably, terms such as "spinal flexion" or "movements of the torso" were selected as an illustrative comparison to enable readers to sufficiently understand the movement dynamics of the robot. The information regarding these motions was not disclosed to the participants, and we did not investigate their specific interpretations of these motions. Our primary objective was to investigate the psychological impact of various movements of the edible robot, rather than replicating human movement. In addition, for the air supply and exhaust cycle, we set up three conditions: short, middle, and long (see Table 4 for each condition). We considered that the speed of the movements could express the difference in alertness, with fast movements expressing excitement and slow movements expressing relaxation. Notably, fast movements are not inherently lifelike. However, we hypothesize

**Table 4. Patterns of air supply and exhaust for the operation of the edible robot.**

| Label | | Cycle of air supply and exhaust | | |
|---|---|---|---|---|
| | | **Short** | **Middle** | **Long** |
| Timing of air supply and exhaust | Alternating | *AS* (Alternating–Short) condition 0.5 s Supply: 0.3 s Exhaust: 0.2 s | *AM* (Alternating–Middle) condition 1 s Supply: 0.3 s Exhaust: 0.7 s | *AL* (Alternating–Long) condition 7 s Supply: 0.3 s Exhaust: 6.7 s |
| | Simultaneous | *SS* (Simultaneous–Short) condition 1 s Supply: 0.3 s Exhaust: 0.7 s | *SM* (Simultaneous–Middle) condition 2 s Supply: 0.3 s Exhaust: 1.7 s | *SL* (Simultaneous–Long) condition 7 s Supply: 0.3 s Exhaust: 6.7 s |

that exceedingly slow motions might undermine the perception of the robot's animateness. Six conditions were set up as aforementioned, and the participants were involved in all conditions.

**Question items.** To investigate perception and taste change only by watching the movements of the edible robot, participants were asked to respond to some items. For each of the six videos, the participants were asked to respond to the following eight items based on a seven-point Likert scale. We adopted the Likert scale methodology to quantitatively assess shifts in specific impressions and emotions. The following are the questionnaire items.

- Do you feel the object's animateness in the video?

- Do you think the object in the video has emotions?

- Do you think the object in the video has intelligence?

- Do you think the object in the video has a hard texture?

- Do you think the object in the video is fresh?

- Do you think the object in the video looks tasty?

- Do you want to eat the object in the video?

- Do you think eating the object in the video is against the moral code?

We used the term "object" instead of "edible robot" in the questions because the term "edible robots" could introduce a bias in the participants' answers. Notably, although our methodology was thematically influenced by Takahashi et al. [21], it did not involve a verbatim replication of their survey items. Our questions were uniquely devised to suit the specific contours of our research, drawing on broad perceptual themes elucidated in previous works. We acknowledge the imperative for further refinement and rigorous validation of this questionnaire in future research, ensuring its reliability and accuracy in teasing apart these sophisticated constructs. In addition to the eight separate Likert-scale questions, we asked the respondents to write freely about two other items: "What impression did you have of the object's movements?" and "What do you think it tastes like?"

**Procedure.** Participants who enrolled in Experiment 1 were presented with a 15-s video showing the movements of the object (the edible robot). They answered 10 questions regarding their impression of the movements of the object. Participants watched the videos of all six conditions. The order of these videos was randomized across participants to address order effects.

**Table 5. Pattern matrix of Experiment 1.**

| | Factor | |
| --- | --- | --- |
| | **1** | **2** |
| **Perception** ($\alpha$ = 0.917) | | |
| Emotion | 0.988 | −0.048 |
| Animateness | 0.840 | 0.040 |
| Intelligence | 0.813 | 0.057 |
| **(not good) Taste** ($\alpha$ = 0.925) | | |
| Do you think the object in the video looks tasty? | −0.005 | 0.954 |
| Do you want to eat the object in the video? | 0.032 | 0.888 |

Factor extraction method: Maximum Likelihood Method

Rotation method: Promax method with Kaiser's normalization

## Results

Initially, an exploratory factor analysis was performed on the eight items from the Likert scale questionnaire. The analysis yielded initial eigenvalues of 4.096 and 1.274 for the first and second factors, respectively, resulting in the extraction of two factors. Subsequent factor analysis was conducted using the maximum likelihood method complemented by a Promax rotation. Within this analysis, items with factor loadings less than 0.50 were eliminated, with three items (hard texture, fresh, and moral) being excluded. The detailed outcomes are presented in Table 5. The cumulative variance explanatory ratio for the two factors was 82.21%. Factor 1 represents the "perception ($\alpha$ = 0.917)"of the edible robot, and factor 2 represents "taste ($\alpha$ = 0.925)".

Further, we executed a within-participant two-factor analysis of variance (ANOVA) using the factor scores from the aforementioned two factors as dependent variables. The independent variables in this analysis were the timing of air supply and exhaust (either simultaneous or alternating) and the cycle duration of air supply and exhaust for the two air chambers (short, middle, or long). The results of the six types of movement on "perception" are shown in Fig 7, and the results on "taste" are shown in Fig 8. To control for Type I errors, alpha values were applied with the Bonferroni correction.

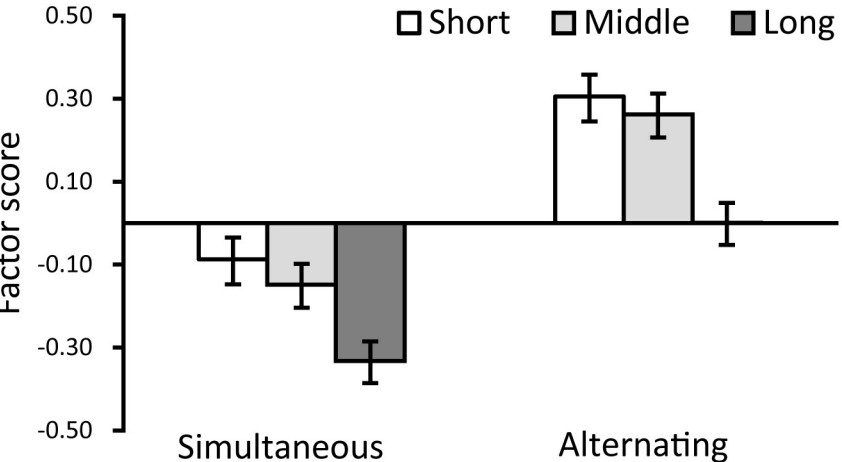

**Fig 7. Results of Experiment 1: Factor 1 (perception).**

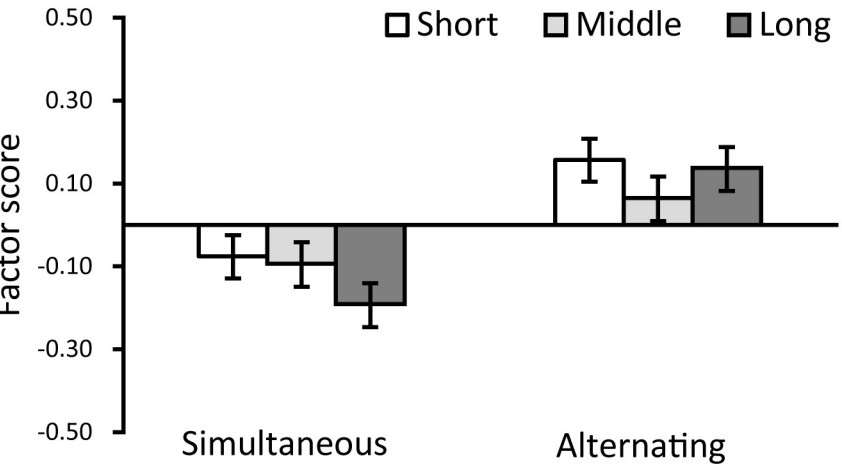

**Fig 8. Results of Experiment 1: Factor 2 (taste).**

Regarding the timing of air supply and exhaust, the results showed that the main effects were significant for both two dependent variables of perception and taste (in order; $F(1, 314) = 135.518$, $p < 0.001$, partial $\eta^2 = 0.301$, $F(1, 314) = 52.825$, $p < 0.001$, partial $\eta^2 = 0.144$). This implies that the robot in the alternating condition yielded an impression of higher perception and positive taste than that in the simultaneous condition. Furthermore, the main effects of the cycle of air supply and exhaust (short, middle, and long) were significant for only the factor of perception ($F(2, 628) = 41.625$, $p < 0.001$, partial $\eta^2 = 0.117$). Multiple comparisons indicated that the scores were significantly higher for "short/middle" than for "long" ($ps < 0.001$). This result suggests that rapid movements might cause the participants to feel that the robot possesses perceptual abilities. Moreover, the interaction effects of air supply and exhaust timing (simultaneous and alternating) and the cycle of air supply and exhaust (short, middle, and long) were significant for only the factor of taste ($F(2, 628) = 4.180$, $p < 0.05$, partial $\eta^2 = 0.130$). The results of a simple main effect test showed that the scores were significantly higher for "short/middle" than for "long" in the simultaneous condition ($ps < 0.05$). This result suggested that slow movements might yield a negative impression regarding taste in the simultaneous condition. However, no such results were found in the alternating conditions.

Furthermore, the results of the text mining analysis on the free-form description of the participants are presented in Tables 6 and 7. Notably, the numbers in the tables indicate Jaccard coefficients [22] for the strength of word-to-word co-occurrence. Regarding the impression of robot movement, participants in the simultaneous condition described the impression as

**Table 6. Free-form descriptions: What impression did you have of the object's movements?**

| Simultaneous | Jaccard coefficient | Alternating | Jaccard coefficient |
|---|---|---|---|
| Hard | 0.068 | Soft | 0.079 |
| Impression | 0.057 | Disgusting | 0.055 |
| Movement | 0.055 | Intense | 0.054 |
| Shaking | 0.040 | Not particularly | 0.051 |
| Stiff | 0.036 | Pulling | 0.047 |

Note: Showing the top five words for each condition.

**Table 7. Free-form descriptions: What do you think it tastes like?**

| Simultaneous | Jaccard coefficient | Alternating | Jaccard coefficient |
|---|---|---|---|
| Tasteless | 0.090 | Sweet | 0.131 |
| Orange | 0.023 | Jelly | 0.056 |
| Bitter | 0.022 | Delicious | 0.026 |
| Pasta | 0.021 | Gummi | 0.023 |
| Salty | 0.020 | Apple | 0.023 |

Note: Showing the top five words for each condition.

"hard" and "shaking," whereas those in the alternating condition, exhibited "soft" and "disgusting" impressions. In addition, regarding the taste of the robot, "tasteless" and "bitter" were reported in the simultaneous condition and "sweet" and "jelly" impressions were reported in the alternating condition. However, co-occurrence was low across all conditions. These results showed that the alternating condition could yield a more tasty and soft-looking appearance than the simultaneous condition.

Based on the results mentioned thus far, for Experiment 2, where we investigated the impressions of the participants after they had eaten the robot, we used the "alternating" condition for the timing of air supply and exhaust. For the cycle duration of air supply and exhaust, we employed the "middle" condition, which was moderate in speed and did not yield a negative impression.

## Experiment 2

Using the movements identified in Experiment 1, in Experiment 2, we investigated the types of impressions participants had when they ate a robot that moved.

### Participants

The experiment was conducted using students at Osaka University, Japan. The participants comprised 10 males and 6 females (mean age = 20.88 years old, range = 19–25 years old, and SD = 1.59 years old). Notably, the participants in Experiment 2 were distinct and did not overlap with those from Experiment 1. All participants were Japanese. We specifically chose Japanese participants because of the cultural influences on the use of onomatopoeic terms, which will be elaborated in a subsequent section. Our major concern was whether cultural background could significantly affect the evaluation using onomatopoeia.

### Materials and methods

This study protocol was approved by the Ethics Committee of the Graduate School of Engineering Science at Osaka University (No. R3–18). All participants provided written informed consent before they participated in the experiment. The experiment was performed per the principles and guidelines of the Declaration of Helsinki (1975).

**Robot movement conditions.** The participants involved in two conditions: stationary and movement. Based on the results of Experiment 1, the movement condition used was the middle cycle of the alternating condition (*AM* conditions in Table 4).

**Question items.** To examine whether impressions such as "perception" and "taste" for eating the edible robot changed in response to the robot's movements, the participants were asked to answer the following 13 Likert-scale question items for all stationary and movement

conditions. They were asked to the following 13 items (some items are the same as in Experiment 1) using the seven-point Likert scale for each of the stationary and movement conditions. We adopted the Likert scale methodology to quantitatively assess shifts in specific impressions and emotions. Incidentally, in Experiment 1, we used the index of "morality," but in Experiment 2, we revised it to "guilt," a question item that is likely to be linked to a specific action, to evaluate the impression of eating a robot.

- Did you think what you just ate had animateness?

- Did you feel an emotion in what you just ate?

- Did you think what you just ate had intelligence?

- Did you think what you just ate had a hard texture?

- Did you feel freshness in what you just ate?

- Did you think what you just ate tasted good?

- Did you feel guilty about what you just ate?

- Did you taste sweetness in what you just ate?

- Did you taste sourness in what you just ate?

- Did you taste bitterness in what you just ate?

- Did you taste spiciness in what you just ate?

- Did you taste saltiness in what you just ate?

- Did you taste umami in what you just ate?

In addition to the 13 separate Likert-scale questions, 14 onomatopoeia (e.g., Gabu (Grappling), Kori-Kori (Crisp)) related to eating were extracted from the Japanese onomatopoeia dictionary [23]. The participants were asked to respond to the texture of what they just ate with the corresponding onomatopoeia (multiple responses allowed). Onomatopoeia is effective in expressing sensations such as texture that cannot easily be quantified. For example, onomatopoeia has been reported to be effective in conveying the sensation of symptoms of illness [24]. Moreover it has been reported to be effective in conveying food impressions [25]. Unlike scales that request impressions with numerical values, onomatopoeia is superior in that it can express differences in quality that cannot be captured with mere numerical values. Moreover, the participants were asked to respond freely in writing the following three questions: a) "What was your impression of what you just ate?," b) "What did it taste or smell like?," and c) "What food ingredients do you think what you just ate is made of?" As the reference to an "edible robot" may introduce a bias in the participants' answers, they were told that the edible robot was "a stick-like object that can be eaten."

**Procedure.** Participants, initially briefed in the front chamber, were then moved to a laboratory where an edible robot was placed on a desk. They took a seat in front of the robot. After 15 seconds, the participants held the 3-D printed handle and lifted the edible robot with their right hands. Then they kept the edible part in their mouths for 10 seconds. The reason for keeping the robot in their mouths for 10 seconds was to ensure that participants fully recognized the sensation of the robot moving inside their mouths under the moving condition. After 10 seconds, they were asked to bite off and eat the edible robot. Those who could not swallow the edible robot were instructed to spit it out. Participants received instructions remotely via a microphone in a separate room. After eating the robot, the participants were

moved to the front chamber and instructed to answer questions on their impressions of the edible robot using a personal computer. After answering the questions, the participants rinsed their mouths with water and involved in a different condition. The order in which the participants were involved in the two conditions (the stationary and movement conditions) was counterbalanced. The duration of the experiment was approximately 45 minutes and each participant was paid 1300 yen.

## Results

In Experiment 2, to investigate in detail the impression of eating the robot, we added more items than those used in Experiment 1. Consequently, we conducted an exploratory factor analysis on these 13 items. The analysis yielded initial eigenvalues over 1.0 of 3.859, 2.440, 1.988, and 1.248, resulting in the extraction of four factors. Subsequent factor analysis was conducted using the maximum likelihood method complemented by a Promax rotation. Within this analysis, items with factor loadings less than 0.50 of each factor were eliminated, with five items (hard texture, tasted good, sweetness, saltiness, and umami) excluded. Thus, two factors were extracted. The detailed outcomes are presented in Table 8. The cumulative variance explanatory ratio for the two factors was 59.665%. Factor 1 represents the "perception ($\alpha$ = 0.858)" of the edible robot, and factor 2 represents "(not good) taste ($\alpha$ = 0.661)".

Subsequently, using the two extracted factors as dependent variables, we investigated the impression of eating a moving robot by comparing the stationary and movement conditions. Given the confirmed normality for both perception- and taste-dependent variables, we conducted a corresponding t-test. Moreover, alpha values were adjusted with the Bonferroni correction. The analysis exhibited a significant difference in the perception of the robot ($t(15)$ = 5.245, $p < 0.001$, $d = 1.311$, 95CI% = 0.637–1.510). This result indicated that the movement condition was significantly higher than the stationary condition. This implied that when the participants ate a moving robot, they could easily perceive the robot as having the ability to be perceptive. However, no significant differences was observed between the conditions for the dependent variable of taste ($t(15)$ = 1.195, $ns$, $d = 0.299$, 95CI% = −0.091–0.325). This implied that either they were moving or did not affect the sense of (not good) taste.

**Table 8. Pattern matrix of Experiment 2.**

| | Factor | |
|---|---|---|
| | 1 | 2 |
| **Perception** ($\alpha$ = 0.858) | | |
| Intelligence | 0.921 | −0.103 |
| Emotion | 0.898 | −0.117 |
| Animateness | 0.662 | 0.070 |
| Guilty | 0.650 | 0.192 |
| Freshness | 0.588 | 0.052 |
| **Taste** ($\alpha$ = 0.661) | | |
| Sourness | −0.163 | 0.842 |
| Spiciness | 0.040 | 0.737 |
| Bitterness | 0.211 | 0.708 |

Factor extraction method: Maximum likelihood method

Rotation method: Promax method with Kaiser's normalization

In addition, Tables 9–11 show the results of the responses to the questions "What was your impression of what you just ate?" "What did it taste or smell like?" and "What food ingredients do you think what you just ate is made of?" These tables list the top five words for each of the stationary and movement conditions. Notably, the numbers in the tables indicate Jaccard coefficients for the strength of word-to-word co-occurrence. In the stationary condition, participants perceived the robot as "food," whereas in the movement condition, they perceived it as a "creature." These results indicated that the edible robots could cause the participants to feel that they were eating animated beings in the movement condition. Additionally, in the evaluation of the taste of the robot, the participants frequently described it as "juice" and "fruit" in the stationary condition. By contrast, in the movement condition, they frequently described it as "sweet" and mentioned specific tastes such as "apple." Moreover, in the stationary condition, the participants believed that the robot comprised "gelatin" and "sugar," but in the movement condition they believed it comprised "agar" and "fruit juice". Therefore, they believed that the robot comprised different materials under different conditions.

**Table 9. Free-form descriptions: What was your impression of what you just ate?**

| Stationary | Jaccard coefficient | Movement | Jaccard coefficient |
|---|---|---|---|
| Eating | 0.250 | Movement | 0.242 |
| None in particular | 0.138 | Creature | 0.161 |
| Food | 0.129 | Mouth | 0.147 |
| Jelly | 0.103 | Disgusting | 0.125 |
| Delicious | 0.100 | Bite | 0.094 |

Note: Showing the top five words for each condition.

**Table 10. Free-form descriptions: What did it taste or smell like?**

| Stationary | Jaccard coefficient | Movement | Jaccard coefficient |
|---|---|---|---|
| Smell | 0.237 | Sweet | 0.161 |
| Juice | 0.100 | None in particular | 0.129 |
| Eat | 0.100 | Apple | 0.121 |
| Fruit | 0.094 | Juice | 0.069 |
| Sweets | 0.067 | Flavor | 0.069 |

Note: Showing the top five words for each condition.

**Table 11. Free-form descriptions: What food ingredients do you think what you just ate is made of?**

| Stationary | Jaccard coefficient | Movement | Jaccard coefficient |
|---|---|---|---|
| Gelatin | 0.419 | Agar | 0.231 |
| Sugar | 0.192 | Fruit juice | 0.217 |
| Water | 0.167 | Sweetness | 0.182 |
| Apple | 0.160 | Artificial | 0.095 |
| Juice | 0.130 | Taste | 0.095 |

Note: Showing the top five words for each condition.

**Table 12. Number of onomatopoeia that appeared for each condition.**

| Onomatopoeia | | Condition | |
|---|---|---|---|
| **Japanese** | **English** | **Stationary** | **Movement** |
| Kori-Kori | Crisp | 7 | 8 |
| Gabu | Grappling | 4 | 8 |
| Munya-Munya | Mumble | 8 | 1 |
| Nicha-Nicha | Sticky | 3 | 7 |
| Guchu | Sopping | 4 | 6 |
| Kucha-Kucha | Crunching | 4 | 6 |
| Gappuri | Firmly (grasped) | 3 | 4 |
| Musha-Musha | Munching | 1 | 2 |
| Picha-Picha | Splashing | 1 | 1 |
| Bori-Bori | Munching | 0 | 1 |
| Gatsu-Gatsu | Gobbling | 0 | 0 |
| Gari-Gari | Crunchy | 0 | 0 |
| Kari-Kari | Crunchy | 0 | 0 |
| Kusha-Kusha | Crunchy | 0 | 0 |

Subsequently, we examined whether the texture of the edible robot when bitten and chewed differed between the tested conditions. For texture, the participants were asked to multiple select the corresponding onomatopoeia. Table 12 presents the onomatopoeia selected by the participants in each condition. In the stationary condition, a significant difference in the onomatopoeia type (Cochran's $Q(13) = 42.23$, $p < 0.001$) was observed. Adjusted multiple comparisons indicated that "Munya-Munya (mumble)" was selected significantly more often than Musha-Musha (Munching), Picha-Picha (Splashing), Bori-Bori (Munching), Gatsu-Gatsu (Gobbling), Gari-Gari (Crunchy), Kari-Kari (Crunchy), and Kusha-Kusha (Crunchy; $p < 0.05$ for all pairs). However, in the movement condition, a significant difference in the onomatopoeia type (Cochran's $Q(13) = 48.07$, $p < 0.001$) was observed. Adjusted multiple comparisons indicated that "Kori-Kori (Crisp)" and "Gabu (Grappling)" were selected trend of more often than Bori-Bori (Munching), Gatsu-Gatsu (Gobbling), Gari-Gari (Crunchy), Kari-Kari (Crunchy), and Kusha-Kusha (Crunchy; $p < 0.10$ for all pairs).

## Discussion

This study experimentally investigated human impressions after eating the movable parts of an edible robot developed using gelatin and sugar. To realize the edible moving part, we used pneumatic forces to power the robot's movement. A pneumatically driven robot was developed by creating a balloon structure ("Structure" section). The deformation generated by supplying air inside the balloon resulted in robotic motion. A material that does not break, even when air is supplied, should be used. However, if the strength of the material is increased, it might not be easily bitten and chewed. The jelly made from gelatin can be strengthened by drying it to reduce the water content. However, in this study, it was not dried; rather, calcium carbonate was added to the material to enhance its strength ("Edible material" section). The tensile-strength test ("Tensile-strength testing" section) confirmed that the elongation and strength of the material increased with the addition of calcium carbonate. We measured the hardness of the material ("Hardness measurements" section) and fabricated it such that it could be driven by pressurized air without breakage. Notably, this hardness was similar to that

of commercially available gummies. We created a metal mold to facilitate the massive production of edible robotic parts.

The experiments investigated the influence of the movements of the edible robot on participants' impressions, particularly regarding their perception of the edible robot and its taste. In Experiment 1, we assessed the participants' impressions on different robot movements through an online survey. Our findings revealed that alternating movements were associated with higher perception and taste factor impression toward the edible robot than simultaneous movements. Additionally, middle- and short-cycle movements exhibited a higher perception impression of the edible robot than long-cycle movements. Therefore, in Experiment 2, we implemented the middle cycle of alternating movements in an edible robot and examined the impressions of the participants when consuming a moving robot. The results indicated that their perception of the edible robot was influenced by its movement, with the moving robot eliciting different impressions compared with the stationary robot. Moreover, a noticeable difference in the perceived texture emerged when participants bit and chewed on the robot under two distinct conditions. When biting and chewing a moving robot, they consistently reported experiencing specific textures (onomatopoeia): "Kori-Kori (Crisp)" and "Gabu (Grappling)." Conversely, when the robot was not in motion, the participants typically perceived a different texture (onomatopoeia): "Munya-Munya (Mumble)." Through this approach, we scrutinized the tactile sensations of eating in detail using onomatopoeia as a measure. Results revealed that despite being artificial, our robot movements could create different textures. Our study results suggest that the incorporation of specific movements into the robot can cause the participants to generate a high perceptions toward the robot. A possible interpretation is that when consuming the edible robot, the participants experienced heightened perceptions because they attributed lifelike qualities such as emotion, intelligence, and animateness to the robot. The utilization of edible robots in this study enabled us to examine the effects of subtle movement variations in human eating behavior under controlled conditions, a task that would be challenging to accomplish with real organisms.

## Limitations

This study has the following limitations. The movements of the edible robot were not designed to convey a specific intent or characteristic but were determined based on what the developed robot could realistically achieve. However, participants might have attributed meanings to the movements of the robot that were not intended by us.

In Experiment 2, the time required for participants to bite and chew the robot following the instruction could have served as an objective measure that could complement the subjective questionnaires. As the robots were powered by air pressure, the onset of a decrease in air pressure inside the air chamber when biting could be used to measure the time from placing the robot in the mouth to the start of biting. In future studies, we will address this limitation by incorporating objective evaluations based on behavior, thus further enhancing the validity of our findings.

We designed the color and appearance of the robot as simple and featureless to prevent those factors from affecting the evaluation. If the experience of eating a living creature is intended to be realized using the edible robot, the robot should mimic a living creature. Furthermore, the motion of the robot could trigger the social experiences of the participants [26–28] and influence the results. Thus, the motion should be effectively designed. An alternative would be to blindfold the participant so that the appearance of the robot does not affect the evaluation during the experiment.

Although the edible component is a pivotal tool for this study, it lacked autonomy in its functionality. For the robot to move, it should be connected to the base component for air supply. This design prerequisite restricted our presentation options for consumption. Specifically, serving only the edible part on a plate without its essential base was impractical. Considering the unwieldy nature of the base component, we adopted a practical solution by encasing it within a cup-like holder equipped with a handle for easier manipulation. However, this adaptation inadvertently introduces an uncertainty, the potential influence of the holder on the participants' perception of the edible robot. The current experimental design could not account for or assess the potential perceptual implications cause by the holder. Moreover, the study design required that participants should hold the robot in their mouth for 10 s to experience its movement. Although this was vital for our primary objective to compare the perceptions between a moving and stationary robot, it did not permit other consumption methods, such as immediate biting, which might elicit different perceptions. In future studies, we will investigate alternative consumption methods and their effects on perception. Furthermore, we will develop the edible parts of the edible robot to operate independently from the other components.

Our proposed edible robot design does not specifically mimic any particular biological form. To address these limitations, we will focus on the field by designing edible robots that imitate forms relevant to ongoing discussions on food shortages and cultural delicacies. Specifically, in future studies, we will emulate creatures consumed in contexts such as insect-based diets, which are being considered as a solution to food scarcity issues, and traditional Japanese dishes like "Odorigui" or "Ikizukuri (live fish sashimi)." These imitations are expected to provide deep insights into the psychological and cognitive responses elicited when consuming moving robots, merging technology with necessities and culinary traditions.

Additionally, we acknowledge the inherent challenges associated with the within-participants experimental design used in Experiment 2. This design can introduce order effects, even if conditions are counterbalanced. For instance, after experiencing a robot that moves, subsequent exposure to a robot that does not move might bias the participant to perceive it as "dead". We selected a within-participant design in Experiment 2 because of time constraints. Nevertheless, these biases might exist, and we believe that future studies would benefit from inter-participant comparisons to mitigate such order effects. Additionally, another significant limitation is the small number of participants, with only 16 participants in Experiment 2. In the analysis of Experiment 2, significant differences were observed, and with the power of the test yielding a high value, it is likely considered sufficient to detect differences between conditions. However, the limited sample size could impact the generalizability of our findings, possibly increasing the risk of type II errors. Future replications of this experiment should consider enlarging the sample size to ensure robust and reliable outcomes.

In addition, our data can be further enriched through qualitative insights. Employing semi-structured interviews in conjunction with questionnaires will enhance the understanding of the participants' impressions of edible robots. This approach could reveal subjective experiences beyond what standard surveys can capture, emphasizing the depth and variety within individual responses.

## Future direction

The following are considered for future studies on HERI. For example, investigations on the cultural differences in the impressions of eating robots in motion, creating new eating experiences using edible robots, and providing entertaining meals. Medical applications, such as brain activation via oral stimulation [29, 30] using edible robots, are also important. Studies on HERI could accumulate knowledge about the effects of eating robots on human psychology

and the modulation of these effects, which are considered to occur when edible robots are used for future medical treatment. To this end, we will develop an edible part that can perform additional complex movements that will possess miniaturized driving capabilities and control devices. Recent 3-D printing technology [9, 31] can be used to create a complicated structure. We will also evaluate the impressions caused by differences in shape and movement.

Finally, we believe that studies on eating-enabled robots are relevant because they stimulate discussion in various interdisciplinary research areas. For example, academic contacts with the humanities are interesting. In many cases, humans prekill food creatures and subsequently cook and eat them. Currently, the presence of these living animals is significantly reduced when food arrives at the table. However, humans are also animals, as many animals feed themselves by eating living creatures. In other words, our eating behavior is a combination of cultural, social, and animal factors. In particular, the animal component of human-eating behavior, including the effect of moving entities on our appetites and eating behaviors, should be considered. From this biological point of view, we believe that the developed edible robot, which can control its movements and sense of taste, shows great potential for the study on eating behavior. Furthermore, our robot may be useful in cultural anthropology to examine the combination of such biological human elements with cultural elements for creating a food culture and identifying the minimalistic elements of this culture. As aforementioned, our robot is expected to be an effective tool for the integrated exploration of the humanities and social sciences, biology, and psychology of eating behaviors.

The development of a robot that appears to be alive and the experience of eating it can lead to food education. We expect that the broadening of philosophical discussions on topics such as what life is and the emotions that arise when robots are eaten, as these are modulated by the developed robot in this study, will lead to a sufficient understanding of bioethics and stimulate moral education efforts.

In conventional studies on human–robot interactions, the impression concerning the robot was often ascertained by participants in the form of questionnaire ratings. These subjective methods did not sufficiently reflect unconscious human processes. We believe that providing users with the experience of "eating a robot," shows great potential, as this may highlight subconscious human attitudes, which cannot be quantified based on questionnaires, toward living organisms and robots. We believe that eating robots will be an attractive subject for stimulating interdisciplinary research discussions and developing discussions involving various studies in the future.

## Conclusion

In this study, we experimentally investigated human impressions associated with the visualization of the movements and the eating of edible robots. Although we evaluated the impressions of participants who observed moving robots, their feelings of the animateness of the robot were stronger for lateral vibration than vertical vibration. Notably, no difference in taste was reported when the robot was moving compared with when it was stationary. However, different perceptions were reported when they ate robots in motion. In addition, when they described the texture using onomatopoeia, specific ones were used for robots in motion compared with stationary ones. Our findings showed that humans perceive specific impressions when eating moving objects. The experimental results provide insights into the effect of the movements of a robot on humans in practical applications of edible robots.

## Supporting information

**S1 Fig. Preparation process for the edible parts.** The process was performed in the order from (a) to (i). The edible parts were made in a kitchen and inspected by the Public Health

Department. (a) Preparation of 100 mL of apple juice, (b) addition of 30 g of sugar and 1 g of calcium carbonate and mixing, (c) complete dilution of the sugar and calcium carbonate by heating, (d) addition of 25 g of gelatin, and (e) stirring until gelatin is completely dissolved. Removal from heat and allow slightly cooling. (f) During cooling, assemble the mold (fixing using a rubber band to prevent the mold from shifting). (g) Pouring of the mixed materials into the mold, (h) insertion of the mold to form the chambers of the edible part, and (i) refrigeration for half a day.
(TIF)

## Acknowledgments

The authors express their gratitude to Professor Aiko Tanaka for valuable discussions and Mr. Hiroaki Ukita, the owner and Chef of the French restaurant "Franc et élégan" for his support in preparing the edible part of the robot for Experiment 2.

## Author Contributions

**Conceptualization:** Yoshihiro Nakata, Midori Ban, Kazuya Horibe, Hideyuki Takahashi, Hiroshi Ishiguro.

**Data curation:** Midori Ban, Ren Yamaki.

**Formal analysis:** Yoshihiro Nakata, Midori Ban.

**Funding acquisition:** Yoshihiro Nakata, Hiroshi Ishiguro.

**Investigation:** Yoshihiro Nakata, Midori Ban, Ren Yamaki.

**Methodology:** Yoshihiro Nakata, Midori Ban, Kazuya Horibe, Hideyuki Takahashi.

**Project administration:** Yoshihiro Nakata.

**Software:** Ren Yamaki.

**Supervision:** Yoshihiro Nakata, Midori Ban, Kazuya Horibe, Hideyuki Takahashi, Hiroshi Ishiguro.

**Validation:** Yoshihiro Nakata, Midori Ban, Ren Yamaki.

**Visualization:** Yoshihiro Nakata, Ren Yamaki.

**Writing – original draft:** Yoshihiro Nakata, Midori Ban, Ren Yamaki.

**Writing – review & editing:** Yoshihiro Nakata, Midori Ban, Kazuya Horibe, Hideyuki Takahashi.

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
