## [Decision Letter · Decision Letter 0]

12 Mar 2023

PONE-D-23-00718

How Does the Experience of Eating an Edible Robot Impact Our Mind?

PLOS ONE

Dear Dr. Nakata,

Thank you for submitting your manuscript to PLOS ONE. After careful consideration, we feel that it has merit but does not fully meet PLOS ONE’s publication criteria as it currently stands. Therefore, we invite you to submit a revised version of the manuscript that addresses the points raised during the review process.

Please, be sure to express the main idea of the manuscript clearly, detail each supporting experiment performed to prove it, and verify that no potential reader may have any doubt about every information provided and declared in the manuscript. Besides, read carefully the reviewer's comments and answer all of them.

Submit your revised manuscript by Apr 26 2023 11:59PM. If you will need more time than this to complete your revisions, please reply to this message or contact the journal office at plosone@plos.org. Please include the following items when submitting your revised manuscript:

A rebuttal letter that responds to each point raised by the academic editor and reviewers. You should upload this letter as a separate file labeled 'Response to Reviewers'.A marked-up copy of your manuscript that highlights changes made to the original version. You should upload this as a separate file labeled 'Revised Manuscript with Track Changes'.An unmarked version of your revised paper without tracked changes. You should upload this as a separate file labeled 'Manuscript'.

We look forward to receiving your revised manuscript.

Kind regards,

Santiago Casado Rojo, Ph.D.

Academic Editor

PLOS ONE

Journal Requirements:

Reviewers' comments:

Reviewer's Responses to Questions

**Comments to the Author**

1. Is the manuscript technically sound, and do the data support the conclusions?

Reviewer #1: Yes

Reviewer #2: Partly

2. Has the statistical analysis been performed appropriately and rigorously? 

Reviewer #1: Yes

Reviewer #2: Yes

3. Have the authors made all data underlying the findings in their manuscript fully available?

Reviewer #1: Yes

Reviewer #2: No

4. Is the manuscript presented in an intelligible fashion and written in standard English?

Reviewer #1: Yes

Reviewer #2: Yes

5. Review Comments to the Author

Reviewer #1: This paper explores the intriguing experience of eating robots. It evaluates the psychological impact of eating a "leaving" creature. The authors designed an experience that involves two aspects that are not regularly experienced together: watching the creature move (i.e. being alive), and eating it. I find this idea highly interesting and I believe it has a very strong contribution to a wide range of research fields ranging from human-robot interaction to understanding human behavior.

The overall presentation of the paper is good. It is well-written and the arguments are clearly presented. The description of the robot's fabrication process is detailed and clear.

While the paper is highly innovative and interesting, some aspects of the work should be further exlained.

1. I believe that the main contribution of this paper is in evaluating the experience of eating a robot. As such, the design considerations that contribute to the experience should be elaborated. The specific robot appearance and the movements chosen for the first experiment should be justified and an explanation of the automatic social experience triggered when interacting with non-humanoid robots should be presented in order to strengthen the claim that there is a significant difference between eating a moving vs not moving robots (see suggested references at the end of the review).

2. It is important to explain what was the predicted experience for each of the six videos and to translate the technical variables into experience terms. What did the authors expect participants to experience and why the specific technical manipulation of the object is predicted to lead to such experience.

3. In experiment 1 The use of the "object" instead of "robot" is appreciated. However, the authors should explain if the order of the videos was counterbalanced and if they controlled for order effects.

4. It is important to explain why the authors asked about the perceived taste in their survey (experiment 1). These questions may bias the perception of the robot's animacy which was the main purpose of this study

5. It would be better to report the results of Experiment 1 immediately after its method.

6. In experiment 1, it is not clear why each question was analyzed separately. It seems that the questions can be divided into two categories - the perception of the robot as animated and the perception of the robot as food - while it would be better to test the validity and reliability of these two categories, I acknowledge that it is beyond the scope of this paper, but I think each category should be analyzed as a subscale instead of analyzing the answer to each question separately. This is in line with the conclusions that the authors derive for experiment 1 (separately discussing animacy and taste).

7. The authors should also explain how they analyzed the answers to the open questions (thematic analysis?) and if possible provide the percentage of participants that gave the answers reported in the paper.

8. Some statements should be toned down (e,g, "a robot that moves like a living creature")

9. The low number of participants in Experiment 2 should be discussed as a limitation as well as the choice of a within-participants experimental design. Such a within experimental design is subjected to order effects (counterbalancing the conditions does not resolve this since after experiencing a robot that moves, a robot that doesn't move may be perceived as "dead" instead of as an object). This choice should be explained and should be clearly discussed as a limitation of the work.

10. Another choice that should be clarified is the use of onomatopoeia as a measure. Why is it the best fit? Is it a validated measure? What is its contribution over other well-validated measures?

11. I am also wondering why the authors did not conduct an in-depth semi-structured interview. With 16 participants and such an intriguing innovative research question, it would be highly informative to get details about the participants' subjective experience (if analyzed appropriately).

12. Some objective measures would also benefit such a novel exploration (e.g. reaction time measures or coding participants' behavior).

13. The authors should also clarify why participants were asked to hold the object in their mouth before eating it. This is not clear and may bias the experience (from a regular eating experience).

14. The chi-square analysis should be better explained. What were the variables included (not conditions since there is a separate analysis for each condition) and what is onomatopoeia type? This was not defined in the method section.

15. The authors should report exact p values and 0.1 cannot be considered marginally significant.

16. Comments 5-7 about the analysis of Likert and open questions are also relevant to experiment 2.

17. In the discussion the authors should discuss the wide range of implications their study has for the research community, both in terms of psychological experiences and for the HRI community. What can we learn about human nature and eating habits? How does reminding people that their food is/was alive changes their eating experience? For the HRI community, there are great implications for understanding the concept of non-humanoid robots and their perception as leaving creatures. If people feel guilty eating food that has autonomous movement this highlights humans' tendency to animate very simple objects that perform a minimal movement - to the extent that they feel guilty when eating them despite clearly understanding that they are not alive.

In sum, I find this work highly interesting and innovative. I think it can contribute to various research fields including psychology and HRI and encourage a wide range of future studies. At the same time, I encourage the authors to clarify and justify some of the design and analysis choices, add a limitations section, and rewrite the discussion to better represent the profound impact of this interesting work.

Anderson-Bashan, L., Megidish, B., Erel, H., Wald, I., Hoffman, G., Zuckerman, O., & Grishko, A. (2018, August). The greeting machine: an abstract robotic object for opening encounters. In 2018 27th IEEE International Symposium on Robot and Human Interactive Communication (RO-MAN) (pp. 595-602). IEEE.‏

Erel, H., Shem Tov, T., Kessler, Y., & Zuckerman, O. (2019, May). Robots are always social: Robotic movements are automatically interpreted as social cues. In Extended abstracts of the 2019 CHI conference on human factors in computing systems (pp. 1-6).‏

Ju, W., & Takayama, L. (2009). Approachability: How people interpret automatic door movement as gesture. International Journal of Design, 3(2).‏

Reviewer #2: This study aimed to investigate the subjective response of eating an edible artifact. I have the following suggestions.

The abstract should be improved by combining the objectives, short methodology, main findings result, and prospective application.

The title is so fictitious and should be revised and align with novelty of this research.

Please write down the contribution of the study at the end part of the Introduction section in bulleted form.

Authors need to add human experimental protocol and study group size. Need to add IRB if available.

Authors only reported subjective results, may be influenced by bias. Objective results should be more appropriate.

6. PLOS authors have the option to publish the peer review history of their article (what does this mean?). If published, this will include your full peer review and any attached files.

Reviewer #1: **Yes: **Hadas Erel

Reviewer #2: **Yes: **Iqram Hussain

---

## [Author Response · Author response to Decision Letter 0]

20 May 2023

Dear Reviewers,

We wish to resubmit the manuscript titled "Exploring the Eating Experience of a Pneumatically-Driven Edible Robot: Animateness, Guilt, and Texture Perception." The title has been updated in response to the reviewer's comment, and the manuscript ID is PONE-D-23-00718.

We would like to express our deepest gratitude for the time and effort you have committed to reviewing our work for publication in PLOS ONE. Your collective feedback has been invaluable, providing us with insightful and helpful comments that guided us in revising our manuscript. We have made significant changes throughout the document, including an edited title and references, to comprehensively address the reviewers’ concerns.

In our resubmission, we believe that these revisions enhance the conclusions of our paper and we eagerly look forward to your further consideration. We are hopeful that our corrections and responses adequately address the issues raised during the review process. A highlighted revised manuscript is included, and we have attached our responses to the reviews in a separate PDF.

In an effort to improve the manuscript's readability, we have sought the assistance of professional editing services to proofread and polish the English language. We trust that this final version meets your standards and look forward to your feedback on this submission.

Thank you for your consideration, and we eagerly anticipate your response.

Sincerely,

Yoshihiro NAKATA

---

## [Decision Letter · Decision Letter 1]

7 Aug 2023

PONE-D-23-00718R1Exploring the eating experience of a pneumatically driven edible robot: Animateness, guilt, and texture perceptionPLOS ONE

Dear Dr. Nakata,

Thank you for submitting your manuscript to PLOS ONE. After careful consideration, we feel that it has merit but does not fully meet PLOS ONE’s publication criteria as it currently stands. Therefore, we invite you to submit a revised version of the manuscript that addresses the points raised during the review process. At this revision stage of your manuscript, several new comments from different reviewers have emerged. Please, try to reply to all of them (specially to those recommending the rejection of the manuscript) in a way in which no further concerns could arise from this side again.

We look forward to receiving your revised manuscript.

Kind regards,

Santiago Casado Rojo, Ph.D.

Academic Editor

PLOS ONE

Journal Requirements:

Reviewers' comments:

Reviewer's Responses to Questions

**Comments to the Author**

1. If the authors have adequately addressed your comments raised in a previous round of review and you feel that this manuscript is now acceptable for publication, you may indicate that here to bypass the “Comments to the Author” section, enter your conflict of interest statement in the “Confidential to Editor” section, and submit your "Accept" recommendation.

Reviewer #1: (No Response)

Reviewer #3: (No Response)

Reviewer #4: All comments have been addressed

2. Is the manuscript technically sound, and do the data support the conclusions?

Reviewer #1: Yes

Reviewer #3: No

Reviewer #4: (No Response)

3. Has the statistical analysis been performed appropriately and rigorously? 

Reviewer #1: No

Reviewer #3: Yes

Reviewer #4: (No Response)

4. Have the authors made all data underlying the findings in their manuscript fully available?

Reviewer #1: Yes

Reviewer #3: No

Reviewer #4: (No Response)

5. Is the manuscript presented in an intelligible fashion and written in standard English?

Reviewer #1: Yes

Reviewer #3: Yes

Reviewer #4: (No Response)

6. Review Comments to the Author

Reviewer #1: I thank the reviewers for the elaborate comments

While many of my concerns were addressed, I have a few suggestions for the final version of the paper:

1. Comment 2 - Clarify that the gestures were not designed to convey a specific intent or characteristic but rather to signal that the robot is capable of movement and acknowledge that participants may have assigned meaning to the movements that you did not evaluate

2. Comment 4 - add to the limitations that using these pre-tests may have primed these subjects in participants' minds, which could have biased their perception

3. Comment 6 - I appreciate the authors' indication that the 2 subsets are independent; however, my comment concerned their analysis of single items, which is problematic. I suggest grouping the single items into 2 independent subscales and analyzing the 2 subscales instead of single items. By also adding Cronbach alpha analysis for each subscale, the authors will greatly improve the quality of their results.

4. Comment 7 - Without a clear analysis methodology, the qualitative data should be treated with extra caution. This should be clearly stated. In addition, percentages should be added to the word tables.

5. Comment 8 - my comment did not relate to phrasing in general but to strong claims that are not supported by data (see my example in the original comment),

6. Comment 9 - Add to limitations the participants number

7. Comment 15 - Please use the term "trend" instead of "significant trend" - adding the word "significant" is misleading.

8. Comment 16 - Do not treat this finding as an indication that there is no relationship between the variables - Bayesian statistics should be conducted in order to test the lack of effects.

Reviewer #3: I was asked to review the first revision of this manuscript. In doing so, I did not judge whether the revision adequately address the concerns of the first round of reviews.

Introduction

The creation of the HERI research is described as follows:

“On the basis of these insights, we conceived the concept of human–edible robot 13

interaction (HERI) research, which explores a new relationship between humans and 14

robots by mimicking the consumption of living creatures with the use of edible robots 15

that allows for control and regulation of their appearance and movements.”

In order to operate within the HERI framework, the investigators must demonstrate the two essential features of their experimental system:

1. That they can mimic the consumption of living creatures with edible robots.

2. That altering robots’ appearance and movements will alter human behavior.

The investigators state their approach for doing so:

“… we developed edible robots based on considerations of their 27

taste and eating sensations and conducted experiments to investigate the psychological 28

and cognitive effects of the consumption of these moving robots on participants.”

The investigators conducted two studies:

“In a preliminary study, we investigated the visual impression impacted by 33

the movement of the developed edible robot. In the main study, we conducted an 34

experiment with an edible robot under two operating conditions: with and without 35

robotic motion. After parts of the robot were consumed by participants, we evaluated 36

their responses as these pertained to the animateness, deliciousness (taste and texture), 37

appetite, and sense of guilt for eating parts of the robot. These factors are considered 38

essential evaluation items in HERI.”

What is missing from the Introduction, and the paper as a whole, are an explicit set of hypotheses and the predictions which follow from them. Without that framing of the work, this is an exploratory study that is helping to develop and demonstrate the two essential features of the HERI system. Exploratory, developmental studies are important, particularly when developing a new experimental paradigm. Please clarify this.

Edible Robot

The investigators designed and built an edible robot; to be appropriate for the HERI framework, the robot should address the first feature (stated above) needed for HERI. But it is not clear what design specifications the investigators were attempting to achieve. Why this design? From Figure 1, one can see the stick standing vertically. Given that HERI’s goal is to explore the relationship between humans’ eating behavior and edible robots, why, was this design created? Given the stated connection to humans eating live creatures, I was expecting a robot that mimicked a species that humans eat live or at least in a form that suggests to them the live, moving form. For example, worms are a common type of terrestrial and marine animal body form. Other researchers have built pneumatic worms, and the pneumatic stick would seem to be a better animal-like form if it were horizontal, on the table, compared to the stick wiggling in the air. It may be that a vertical stick mimics some live food; but please explain all of these design choices in light of the HERI objectives.

Edible material

Were taste tests conducted on the various materials? How, for example, do tensile stiffness and hardness relate to perceptions of flavor or tastiness? Were these parameters meant to be manipulated as part of the experimental design? If so, why? If not, then why were these mechanical tests included?

Experiment 1

For the robot movement conditions, why compare the motions of the small stick to humans? I don’t think that this is intended, but that description makes me think that you are trying to mimic human motions. This question connects back to my questions about your design target: what, specifically, are you trying to mimic? As for how the vertical stick moves, you can simply say that in one mode it bends side-to-side and in the other it elongates and shortens.

When designing the speed and mode of the movements, here, again, is where reference to real animals that are eaten live – as mentioned in the Introduction – is important. As for the specific durations, why are those specific three levels chosen?

When the questions were first presented, it wasn’t clear that the survey was based on an existing instrument. Then Takahashi et al. [21] was cited after the questions. This is the cited paper:

Takahashi H, Ban M, & Asada M. (2016). Semantic differential scale method can reveal multi-dimensional aspects of mind perception. Frontiers in Psychology. 7:1717.

The problem is that Takahashi et al. (2016) does not contain any survey questions. Instead, that paper cites Gray et al. (2007) and Takahashi et al. (2014), neither of which is cited in this current manuscript. More importantly, the Gray et al. (2007) and Takahashi et al. (2016) papers are about mind perception of a variety of entities, something quite different, I think, to the intention here.

I did more research to see whether I simply hadn’t dug deep enough into the literature. To justify the questions in their questionnaire, Takahashi et al. (2014), cite this paper, which is also not cited in the current manuscript:

Kanda, T., Ishiguro, H., Ono, T., Imai, M., & Nakatsu, R. (2002). An evaluation on interaction between humans and an autonomous robot Robovie. Journal of Robotics Society of Japan,

20(3), 1e9.

I obtained a copy of this manuscript, which is in Japanese. Using ChatGTP, I translated portions of the paper. From that translation, I learned that the work is on a humanoid robot, investigating how movement of that robot impacts psychological reactions of humans interacting with it. This seems even less related to the current project than the work on mind perception.

I’ve been unable to find evidence that the questions in this manuscript have any basis in previous research. Unfortunately, this current survey lacks any basis for validity and reliability. Without validity and reliability, we don’t know what the survey is actually measuring.

Experiment 2

Many of the same problems for Experiment 1’s survey instrument are in play for Experiment 2’s modified version. While I’m intrigued by the use of onomatopoetic terms in describing illness and food impressions, we still have the problem validity and reliability in the current context.

When considering the experimental design, it appears that all participants participated in stationary and animated conditions. What was the nature of the animated condition? Bending or elongation and shortening?

Of more concern is the lack of a set of controls to understand the effects of a vertical stick connected to a cup-like holder relative to the stick unattached that one picks up with one’s fingers. A detached stick would also allow investigation of cutting and utensil use on perception. With controls, one could also seek to understand the effect of having to hold the food in the mouth for 10 seconds compared to just biting, the usual means of eating.

In terms of the results, it seems clear that the two different conditions – stationary or moving – impacted verbal impressions that people gave of what they ate. This is tentative support for the second of the two essential features of the HERI experimental system:

1. That they can mimic the consumption of living creatures with edible robots.

2. That altering robots’ appearance and movements will alter human behavior.

However, the first feature is not addressed in this work.

Reviewer #4: The authors have addressed all the comments from the previous round. Therefore, I am happy to recommend acceptance at this stage.

7. PLOS authors have the option to publish the peer review history of their article (what does this mean?). If published, this will include your full peer review and any attached files.

Reviewer #1: **Yes: **Hadas Erel

Reviewer #3: No

Reviewer #4: No

---

## [Author Response · Author response to Decision Letter 1]

21 Nov 2023

We wish to resubmit the manuscript titled "Exploring the Eating Experience of a Pneumatically Driven Edible Robot: Perception, Taste, and Texture." In the course of additional analyses and a detailed examination of our experimental findings, we recognized the need to accurately represent our research content. Hence, we have modified the title to reflect the depth and nuances of the study. The ID of the manuscript is the ID PONE-D-23-00718R1.

We sincerely appreciate the time and effort you have dedicated to reviewing our work for publication in PLOS ONE. Your collective feedback has provided us with insightful and helpful comments. In response, we have made significant changes throughout the manuscript, including the edited title and updated references to address the reviewers’ concerns. We believe that these revisions have strengthened our conclusions, and we are pleased to resubmit our manuscript for further consideration. We hope our corrections and responses adequately address the concerns raised during the review process, and we eagerly await your feedback on this submission. Please refer to the attached PDF document for detailed responses to each of the reviewers’ comments, with revised portions of the manuscript highlighted for ease of reference. Additionally, a highlighted revised manuscript is included. Finally, we have employed professional editing services to proofread and enhance the English language of our manuscript before resubmission.

While revising our manuscript based on the comments, one of our co-authors, Dr. Midori Ban, made significant additional contributions, especially in conducting the supplemental analyses that were recommended. After a consensus among all co-authors, we decided to change the order of authorship, moving Dr. Midori Ban from the third position to the second. We believe this adjustment reflects the actual contributions to the revised manuscript.

---

## [Decision Letter · Decision Letter 2]

18 Dec 2023

Exploring the eating experience of a pneumatically-driven edible robot: Perception, taste, and texture

PONE-D-23-00718R2

Dear Dr. Nakata,

We’re pleased to inform you that your manuscript has been judged scientifically suitable for publication and will be formally accepted for publication once it meets all outstanding technical requirements.

Kind regards,

Santiago Casado Rojo, Ph.D.

Academic Editor

PLOS ONE

Additional Editor Comments (optional):

Reviewers' comments:

Reviewer's Responses to Questions

**Comments to the Author**

1. If the authors have adequately addressed your comments raised in a previous round of review and you feel that this manuscript is now acceptable for publication, you may indicate that here to bypass the “Comments to the Author” section, enter your conflict of interest statement in the “Confidential to Editor” section, and submit your "Accept" recommendation.

Reviewer #1: All comments have been addressed

2. Is the manuscript technically sound, and do the data support the conclusions?

Reviewer #1: Yes

3. Has the statistical analysis been performed appropriately and rigorously? 

Reviewer #1: Yes

4. Have the authors made all data underlying the findings in their manuscript fully available?

Reviewer #1: (No Response)

5. Is the manuscript presented in an intelligible fashion and written in standard English?

Reviewer #1: Yes

6. Review Comments to the Author

Reviewer #1: I thank the authors for the revision

Most of my concerns were addressed, and I have recommended the acceptance of this very interesting paper.

7. PLOS authors have the option to publish the peer review history of their article (what does this mean?). If published, this will include your full peer review and any attached files.

Reviewer #1: **Yes: **Hadas Erel

---

## [Editor Report · Acceptance letter]

26 Jan 2024

PONE-D-23-00718R2 

PLOS ONE

Dear Dr. Nakata, 

I'm pleased to inform you that your manuscript has been deemed suitable for publication in PLOS ONE. Congratulations! Your manuscript is now being handed over to our production team.

Kind regards, 

on behalf of

Dr. Santiago Casado Rojo 

Academic Editor

PLOS ONE